# An Efficient Clustering Protocol for Cognitive Radio Sensor Networks

**Vladimir Shakhov and Insoo Koo ***

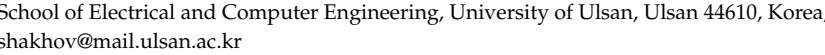

School of Electrical and Computer Engineering, University of Ulsan, Ulsan 44610, Korea; shakhov@mail.ulsan.ac.kr
* Correspondence: iskoo@ulsan.ac.kr

**Abstract:** Wireless sensor networks are considered an integral part of the Internet of Things, which is the focus of research centers and governments around the world. Clustering mechanisms and cognitive radio, in turn, are considered promising wireless network technologies for network management and spectral efficiency, respectively. In this paper, we consider the flaws in the previously proposed network stability-aware clustering technique. In particular, we demonstrate that existing solutions do not operate properly based on the remaining energy and the quality of available common channels, even if their fusion is declared. In addition, security issues have not been sufficiently developed. We offer an approach to address these flaws. To improve protocol efficiency, the problem of parameter tuning is discussed, and a performance analysis of the proposed solution is provided as well.

**Keywords:** cognitive radio; clustering protocol; wireless sensor networks; Internet of Things

## 1. Introduction

Integration of the Internet of Things and wireless sensor networks is paving the way toward optimized production processes, improved operational efficiency in enterprises, and rationalization and delivery of high-quality service. The latest intelligent manufacturing technologies are being developed and implemented around the world. The growth rates in the respective applications have already exceeded the wildest expectations. It is estimated that about 10 billion devices used in the industrial sector are connected to the Internet [1]. However, the rapid expansion of wireless technologies has brought many scientific and technical challenges for both academia and industry. An important and timely problem is the research and development of intelligent and efficient protocols that allow sensor networks to coexist with the existing wireless infrastructure while maintaining the performance required of Internet of Things (IoT) applications [2].

Technologies in cognitive radio (CR) are intended to solve the coexistence problem and improve the fault tolerance of wireless transmissions in a heavily congested environment. In a CR network, a node captures spectrum state information through interference measurements, and then searches for, and uses, the available spectrum resources so that different wireless devices can share the same frequency bands without causing problems for each other. There are two types of user in CR networks. Primary users (PUs) are the licensed users; secondary users (SUs) are allowed to share the licensed channels with PUs provided there is no harmful interference with the PUs. This approach resolves the tension between rapidly growing wireless traffic and spectrum scarcity [3]. Thus, Cognitive Radio Sensor Networks (CRSNs) are capable of meeting the stringent quality of service (QoS) requirements demanded from various IoT applications [4].

To effectively manage communications in a wide variety of distributed wireless systems, a clustering technique is usually used [5]. In accordance with the prescribed rules, neighboring network nodes are combined into groups called clusters. A cluster head is elected from among the cluster members. The cluster head is responsible for intra-cluster communications as well as inter-cluster communications. Clustering protocols are designed

to improve the performance of network communications and ensure stable operation and scalability of the networks. Clustering is very important for CRSNs operating in a highly dynamic, unstable wireless environment due to PU activity. Besides monitoring the geographic proximity and the residual energy of the nodes, clustering protocols for CRSNs have to take into account the common licensed channels available to cluster members. This is called spectrum-aware clustering. To take advantage of clustering, we need to overcome a number of challenges due to dynamic changes in the available channels, the heterogeneous quality of heterogeneous channels, and so on. Thus, research and development of clustering protocols for CRSNs is an important and timely problem. The recent network stability-aware clustering (NSAC) protocol outperforms existing solutions. However, the failures of this protocol motivate us to investigate their causes and, thereby, improve the clustering protocols for CRSN.

The contributions of this paper are summarized as follows.

- We point out critical flaws in the existing network stability-aware clustering technique. We point out that the previously used channel availability metrics are generally untenable. We develop the appropriate formalism to prove this;
- We argue that the clustering procedure has to be revised;
- We discuss how to fix the identified flaws. We offer alternative indicators for the selection of the cluster head. It is also suggested to limit the cluster size to a predetermined number. We present an analytical framework to calculate this number and examine the performance of our proposals. The performance analysis results are provided as well.

The rest of this paper is organized as follows: Section 2 introduces related works. Special attention is paid to a network stability-aware clustering protocol for CRSNs. A critical analysis of the existing NSAC technique is provided in Section 3. Section 4 presents an approach to cluster head selection and cluster formation along with the results of a numerical analysis to evaluate the performance of the proposed method. Finally, Section 5 concludes the paper.

## 2. Preliminaries

### 2.1. Related Works

Clustering is a fairly common technique for cognitive radio networks. Several recent studies have reported that the proper use of this technique can essentially improve the performance of QoS support mechanisms. For example, a Bayesian method for nonparametric channel clustering, which determines the QoS levels supported over the available licensed channels, was proposed [6]. The proposed method is based on an unsupervised clustering scheme and outperforms K-means and other baseline clustering algorithms.

A comprehensive survey of clustering methods for CRSNs was presented [5]. The clustering methods discussed are mainly based on the number of available channels. In addition, the authors of this paper noted that there are not enough clustering investigations to satisfy the particular requirements of CRSNs, such as limited battery power in the sensor nodes and heterogeneous licensed channels.

In [7], the authors modified a basic distributed clustering protocol for wireless sensor networks, named Low Energy Adaptive Clustering Hierarchy (LEACH) [8], and delivered a spectrum-aware extension of the LEACH protocol, named CogLEACH. The modified protocol utilizes the number of free channels as a weight in the probability of each sensor node becoming a cluster head. It was shown that CogLEACH is more efficient than the LEACH protocol. However, the issues of network topology and channel quality were not properly addressed.

In [9], the spectrum-aware clustering approach is based on joint representation of the network topology and spectrum availability in undirected bipartite graphs. To obtain spectrum-aware clusters, the authors offered to solve the problem of constructing bi-cliques of maximum size from the bipartite graphs. The protocol requires heavy computation and ignores the residual energy of the network nodes.

The weighted clustering metric introduced in [10] includes temporal–spatial correlation, confidence level, and residual energy. The authors use a very firm assumption that the Euclidean distance between any two nodes in the network is known and does not change. Moreover, the channel state was ignored.

Recently, the NSAC protocol was offered [11]. Unlike previous protocols, NSAC handles both power consumption and spectrum dynamics simultaneously. The enclosed simulation results demonstrate that NSAC essentially outperforms existent protocols. Let us consider NSAC in detail in the next subsection.

### 2.2. Network Stability-Aware Clustering Protocol

According to the system model [11], a set of licensed channels, $C$, is opportunistically available to the CRSN, and $|C| = m$. A cognitive sensor (CS) may use a licensed channel if it is not used by PUs. PU activity on the $i$th channel is considered a random process with busy and idle states. The probability that the $i$th licensed channel is available to SUs is denoted by $p_i$. Correspondingly, the probability that this channel is used by PUs is $1 - p_i$. The following channel quality metric, $Q_i$, is assigned to each channel $i$:

$$Q_i = (1 + \log_\varepsilon p_i) M_i \tag{1}$$

where $M_i$ is the average idle duration, and $\varepsilon$ is a user-defined parameter.

It is important to note that the authors of NSAC claimed the following instructions regarding the choice of $\varepsilon$:

- $\varepsilon > 1$;
- If $p_i$ is preferable, then select a big $\varepsilon$;
- If $M_i$ is preferable, then select a small $\varepsilon$.

Channel quality metric (1) is used to calculate the weight of a network node in terms of spectrum availability. For this purpose, NSAC uses a graph-theoretic approach similar to the one in [12]. CS $k$ actualizes the sets of its neighboring nodes, $N_k$, and available channels, $C_k$ (i.e., $C_k$ is a subset of $C$), and creates a bipartite graph $(N_k, C_k, L_k)$, where $C_k$ and $N_k$ are independent sets of vertices, and $L_k$ is a set of edges. An edge, $l \in L_k$, connects vertex $v_N \in N_k$ to vertex $v_C \in C_k$ if channel $v_C$ is available to CS $v_N$. The weight of edge $l$ is defined as the quality of the corresponding channel, $v_C$. Next, the maximum edge biclique $(N_k^*, C_k^*, L_k^*)$ is calculated. The weight of CS $k$ in terms of spectrum availability is defined as follows:

$$W_{k,c} = |C_k^*| \sum_{j \in N_k^*} Q_j \tag{2}$$

We illustrate the calculation of $W_{k,C}$ in Figure 1. The indexes of available channels for each CS are given in square brackets.

The total weight of CS $k$ is defined as

$$W_k = \mu W_{k,c} + (1 - \mu) W_{k,e} \tag{3}$$

where

$$W_{k,e} = \frac{E_k}{E_k + \sum_{j \in N_k^*} E_j} \tag{4}$$

in which $\mu$ is introduced as a balance factor between network stability and remaining energy, and $E_j$ designates the residual energy in CS $j$.

The CS with the largest total weight is marked as the cluster head. This node (e.g., CS $k$) and its neighbors, ($N_k^*$), form a cluster and are excluded from further consideration. Other nodes update environmental information and repeat the process. Thus, a good candidate for the cluster head has enough energy and can use a set of lightly loaded channels.

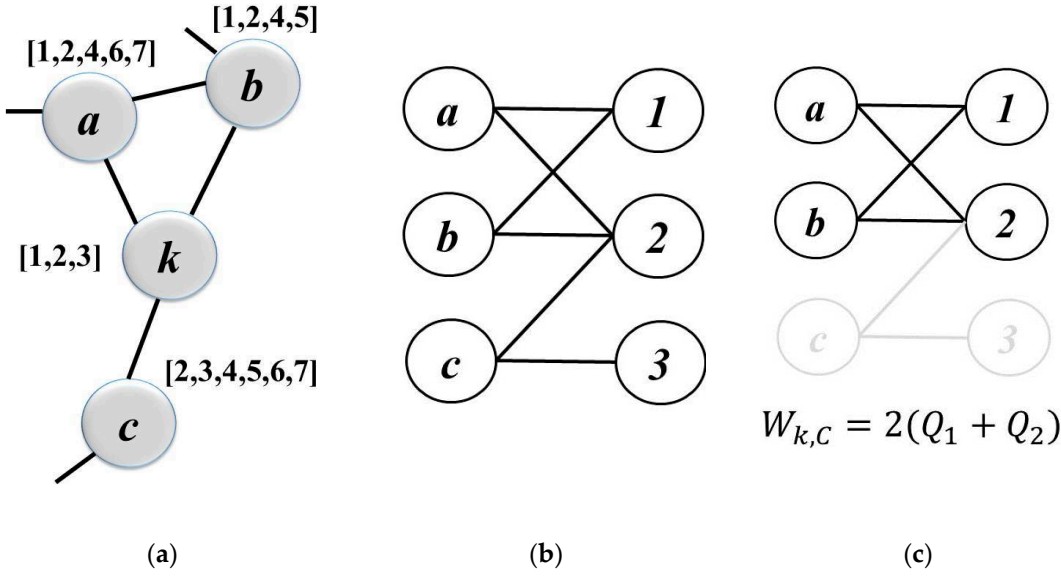

(**a**)                  (**b**)                  (**c**)

**Figure 1.** An example of calculations for the spectrum-aware weight of cognitive sensor (CS) *k*: (**a**) a four-node fragment of the Cognitive Radio Sensor Network (CRSN) topology; (**b**) the bipartite graph for CS *k*; and (**c**) the maximum edge biclique and weight of CS *k* in terms of spectrum availability.

### 3. Critical Analysis of NSAC

In this section, we argue that the NSAC protocol fails to meet the declared goals. Therefore, the protocol as currently presented is impractical for deployment in CRSNs.

#### 3.1. Channel Quality Metric

Let us assume that for some channel $i$ it is observed that $M_i >> M_j$, $p_i \approx p_j$, $\forall j \in C$, and $j \neq i$. The residual energy is the same for all CSs. Obviously, CS $i$ is the best choice for cluster head. However, if we take

$$\varepsilon = \left(\frac{1}{p_i}\right)^2 > 1 \tag{5}$$

then $Q_i = -M_i$ and $Q_i \ll Q_j \forall j$. If we increase the power (making $\varepsilon$ less), then the channel quality will get worse. In Figure 2, we plot the channel quality metric for the different values of $p_i$, $M_i = 1$.

Note that the choice of small $\varepsilon$ (keeping the requirement $\varepsilon > 1$) leads to an unlimited negative value of the channel quality metric for any fixed $M_i$ and $p_i$; i.e.,

$$\lim_{\varepsilon \to 1^+} Q_i = -\infty \tag{6}$$

This contradicts the NSAC authors' assertion regarding the choice of $\varepsilon$. To avoid an absurd situation, the following statement must be true:

$$Q_i(\varepsilon) > 0 \tag{7}$$

so, from (1), we get

$$p_j > \frac{1}{\varepsilon} \ \ \forall j \in C \tag{8}$$

The parameter $\varepsilon$ is universal for all CSs. The rules for applying the clustering protocol are the same throughout the system. Hence, $\varepsilon$ has to be small enough. In fact, it is difficult to maintain inequality (8) due to the high dynamics of the spectrum available to the CRSN and the geographic distribution of its nodes.

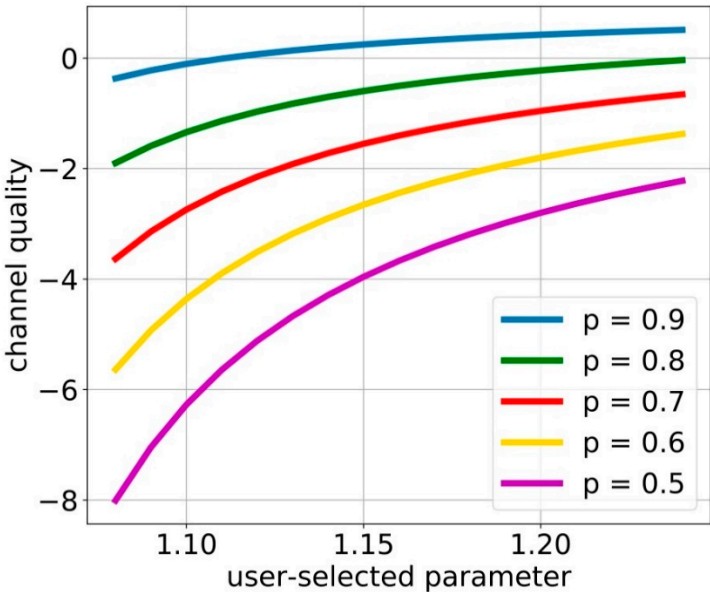

**Figure 2.** Channel quality metric behaviors.

Assume a group of adjacent nodes uses the set of channels $C_0$, $C_0 \subset C$. These CSs exchange information directly, and define $\varepsilon$ such that (8) is true; i.e.,

$$\min_{j \in C_0} p_j > \frac{1}{\varepsilon} \tag{9}$$

If parameter $p_i$. of another channel (i.e., $i \in C \backslash C_0$) has a uniform distribution with CDF

$$U(x) = \begin{cases} 0, & x < a \\ \frac{x-a}{1-a}, & a \leq x < 1 \\ 1, & x \geq 1 \end{cases} \tag{10}$$

where constant $a < \varepsilon^{-1}$, then condition (8) is violated for this channel with the following probability:

$$\frac{\varepsilon^{-1} - a}{1 - a} \tag{11}$$

Therefore, the probability of NSAC protocol failure is as follows:

$$1 - \left( \frac{1 - \varepsilon^{-1}}{1 - a} \right)^{|C \backslash C_0|} \tag{12}$$

For example, if $\varepsilon = 5$, $a = 0.1$, $|C \backslash C_0| = 30$, then NSAC fails with a probability of $\approx 0.97$.

Next, we consider a big $\varepsilon$. Let us take channels $i, j$ such that $M_i = M_j$, $p_j = kp_i$, $1 < k < p_i^{-1}$. It is easy to see that $Q_i < Q_j$. However, for any small $\delta > 0 \; \exists \varepsilon > 1$:

$$Q_j - Q_i < \delta \tag{13}$$

Really,

$$Q_j - Q_i = M_i \log_\varepsilon k \tag{14}$$

Therefore, if

$$\varepsilon > k^{\frac{M_i}{\delta}} \tag{15}$$

then the considered inequality is true. Thus, if $\varepsilon$ is large enough, then $p_i$ becomes unimportant. This refutes the NSAC claims as well.

Parameter $\varepsilon$ is intended to provide a preference between $p_i$ and $M_i$. Let us provide the corresponding formalism.

Define set $C_p$ of channel pairs, as follows:

$$C_p = \left\{ (i,j) \mid i,j \in C, \; M_i < M_j, \; p_i > p_j \right\} \tag{16}$$

We say that $\varepsilon$ delivers the preference for $p_i$ on set $C_p$ (i.e., the probability of channel availability is more important than the average idle duration) if $Q_i(\varepsilon) > Q_j(\varepsilon) \; \forall \, (i,j) \in C_p$.

The provided definition is equivalent to the following condition:

$$\log_\varepsilon \frac{p_i^{M_i}}{p_j^{M_j}} > M_j - M_i \tag{17}$$

Therefore, we obtain

$$\varepsilon < \sqrt[M_j - M_i]{\frac{p_i^{M_i}}{p_j^{M_j}}} \tag{18}$$

which can be rewritten in compact form as:

$$\varepsilon < \delta(i,j) \tag{19}$$

where

$$\delta(i,j) = \frac{1}{p_j} \left( \frac{p_i}{p_j} \right)^\gamma \tag{20}$$

$$\gamma = \frac{M_i}{M_i - M_j} \tag{21}$$

Combining (8) and (19), we obtain a condition for the choice of $\varepsilon$ in the situation when $p_i$ is more important

$$\frac{1}{\min\limits_{i \in C} p_i} < \varepsilon < \min\limits_{(i,j) \in C_p} \delta(i,j) \tag{22}$$

Since $\gamma > 0$, and since $p_i > p_j$, the inequality for the choice of $\varepsilon$ is always compatible. Similarly, to formalize the importance of $M_i$, we define the following set:

$$C_M = \left\{ (i,j) \mid i,j \in C, \; M_i > M_j, \; p_i < p_j \right\} \tag{23}$$

and obtain

$$\varepsilon > \delta(j,i), \; (i,j) \in C_M \tag{24}$$

Combining (8) and (24), we obtain a condition for the choice of $\varepsilon$ when $M_i$ is preferable:

$$\max \left\{ \frac{1}{\min\limits_{i \in C} p_i}; \max\limits_{(i,j) \in C_M} \delta(i,j) \right\} < \varepsilon \tag{25}$$

Thus, the initial single condition for the choice of $\varepsilon$ is not sufficient for the correct functioning of the NSAC protocol.

In short,

1.  NSAC does not provide the proper choice of parameter $\varepsilon$ for the channel quality metric (1);
2.  To use metric (1), NSAC needs to abandon some of the available licensed channels or implement a global mechanism for the dissemination of channel status information;
3.  In general, metric (1) does not automatically provide a tradeoff between the probability of channel availability and the average idle duration in the sense defined above.

### 3.2. CS Weight

In NSAC, the CS weight, $W_k$, has the dimension of time, because the first term, $W_{k,c}$, in Formula (3) has the dimension of time, and the second term, $W_{k,e}$, is normalized and dimensionless. Therefore, the absolute value of $W_k$ is highly dependent on the unit of measurement for idle duration (seconds, milliseconds, years, etc.). In this situation, it is doubtful to get a balance between $W_{k,e}$ and $W_{k,c}$. This is evidenced by the following fact. In simulation experiments in [1], the authors used balance factor $\mu$ as a normalization multiplier: $\mu$ is relatively small, $1 - \mu \approx 1$. So, if $W_{k,e}$ gets an increment comparable to $W_k$ (if compatible), then the new total weight doubles, whereas if $W_{k,c}$ gets an increment of $W_k$, the new weight increases very slightly.

Let us rewrite the residual energy metric in equivalent form:

$$W_{k,e} = \frac{\delta}{\delta + 1} \tag{26}$$

where

$$\delta = \frac{E_k}{\sum_{j \in N_k^*} E_j} \tag{27}$$

Consider a simple case of homogeneous CSs, where each CS has the same residual energy, and all channels are isotropic, i.e., $\left|C_i^*\right| \equiv \left|C_j^*\right|$, $Q_i = Q_j \; \forall i, j \in C$. Only the set of neighbors, $N_k^*$, changes. In this case, $\delta = \left|N_k^*\right|^{-1}$, and the weight of residual energy becomes

$$W_{k,e} = \frac{1}{\left|N_k^*\right| + 1} \tag{28}$$

Clearly, we also have

$$W_{k,c} = |C_k^*| \, Q_1 \, |N_k^*| \tag{29}$$

Therefore, criterion (3) becomes

$$W_k = \mu \, |C_k^*| \, Q_1 |N_k^*| \; + \; \frac{1 - \mu}{\left|N_k^*\right| + 1} \tag{30}$$

Now, it is easy to see that the terms in sum (3) are not balanced. In Figure 3, we show $\mu \, W_{k,c}$ and $(1-\mu)W_{k,e}$ versus $\left|N_k^*\right|$. In this figure, we have taken $\left|C_k^*\right| Q_1 = 10$. By varying $\mu$ from 0.01 to 0.001, we get graphs of three linear functions reflecting the contribution of channel availability to the total weight (the blue lines). The red curve shows the corresponding contribution of residual energy. Here, we present one curve due to the negligible effect of parameter $1 - \mu$.

We can see from Figure 3 that a multiplier is very poor at normalizing factors behind the total weight. Even in a simple case, we cannot guarantee the proper choice for $\mu$. It depends on the dynamics of the system. Thus, the total weight calculation has to be revised.

### 3.3. Clustering Procedure

In the NSAC protocol, in each iteration, one CS with the highest weight is selected as the cluster head. All its neighbors become members of this cluster, regardless of their number. The absence of cluster size control can potentially lead to unsatisfactory quality of service for the cluster members (a high loss rate for internal cluster packets, long packet latency, etc.), and irrational involvement of licensed channels.

Let us consider the CSRN in Figure 4. We assume that the residual energy is the same for each CS. There are three homogeneous licensed channels that CSs are allowed to use. Using the NSAC protocol, we get

$$W_{k,c} = 4q \tag{31}$$

where $q$ is the value of the channel quality metric. Additionally,

$$W_{i,c} = 2q, \ i \in \{a, b, c, d\} \tag{32}$$

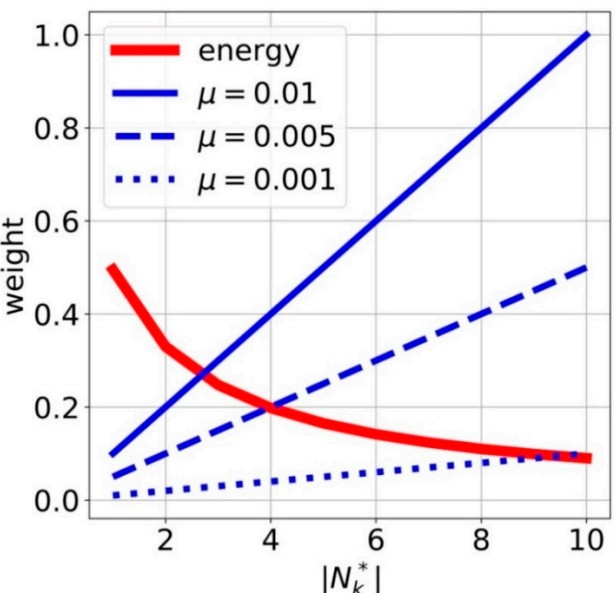

**Figure 3.** The behavior of total weight components.

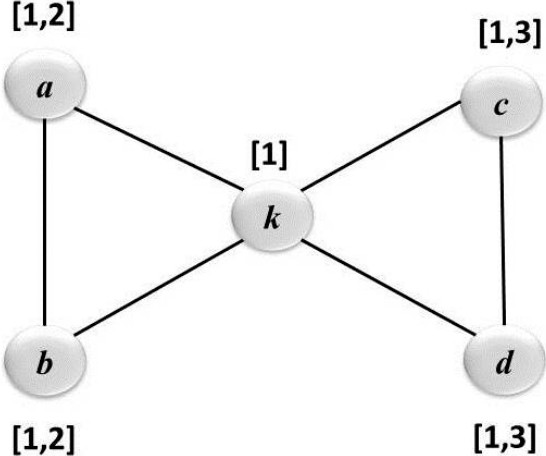

**Figure 4.** An example of irrational involvement of licensed channels.

All nodes form one cluster with CS $k$ as cluster head. In addition, although there are three licensed channels available, only one is used by all SUs, which essentially reduces the QoS. Packet forwarding latency becomes longer due to the increased contention among the CSs.

Furthermore, it should be noted that NSAC, similar to other network protocols with selection of an intermediate node based on self-promotion, is vulnerable to Denial of Service attacks such as Black/Grey Holes. This type of attack is very popular in wireless networks [13–17]. In one scenario for this intrusion, a malicious node acts as a CS and broadcasts an enormous fake weight. As a result, the malicious node is selected as a cluster head for many nodes, and it then destroys the connections. An unlimited cluster size can potentially amplify the effect of this attack.

## 4. Proposition

On the basis of the conducted research, we can conclude that the NSAC protocol, in its present form, might fail to achieve its goals due to the identified critical flaws. In this section, we consider how to fix this. First, we suggest changing the metrics used to select the cluster header. Next, we modify the clustering procedure and calculate the cluster size. To substantiate our proposals, we use models based on continuous time Markov chains. Their use often requires simplifying assumptions. However, these models often provide a basis for adequate conclusions and valuable qualitative results. In particular, Markov process-based methods are generally used when the various performance metrics of wireless communication systems are calculated.

### 4.1. Metrics

First, to resolve the contradictions mentioned above, we propose revising metric (1). Let us consider PU activity on the channels in detail. A channel alternates between idle and busy. To describe the channel status, we use a continuous time, two-state Markov chain, as shown in Figure 5. Similar Markov chain models are usually used in the literature to describe PU activity [9,18].

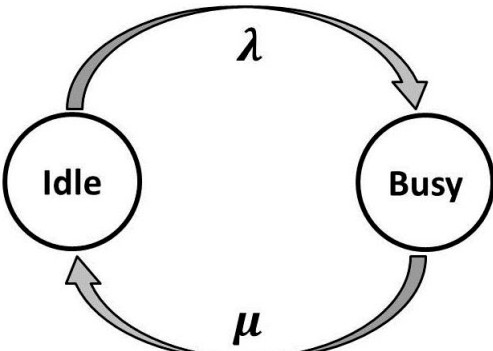

**Figure 5.** Markov model for a licensed channel's status.

This process spends an exponentially distributed amount of time with rate $\lambda$ in the idle state before making a transition to the busy state. Accordingly, the time until a PU releases the channel is an exponential random variable with rate $\mu$.

This is a well-known birth-death process described by a Kolmogorov equation [19]. The vectors of steady-state probabilities are as follows:

$$\left( p_{Idle} , \ p_{Busy} \right) = \left( \frac{\mu}{\lambda + \mu}, \quad \frac{\lambda}{\lambda + \mu} \right) \tag{33}$$

Therefore, for the notation introduced above, we get

$$p_i = \frac{\mu}{\lambda + \mu} \tag{34}$$

and

$$M_i = \frac{1}{\lambda} \tag{35}$$

Note that changing $\mu$ can make significant changes to factor $p_i$, but it has absolutely no effect on $M_i$. At the same time, both $p_i$ and $M_i$ are decreasing by the interval $\lambda > 0$, i.e.,

$$\frac{\partial p_i}{\partial \lambda} < 0; \ \frac{\partial M_i}{\partial \lambda} < 0 \ \ \forall \lambda > 0 \tag{36}$$

Moreover, there is a functional dependency between factors $p_i$ and $M_i$:

$$p_i = \frac{1}{1 + \frac{1}{\mu M_i}} \tag{37}$$

and the inverse dependency:

$$M_i = \frac{p_i}{\mu(1 - p_i)} \tag{38}$$

Thus, from a practical point of view, there is no reason to consider $p_i$ and $M_i$ as competing entities. If $\mu$ is a constant, then $M_i$ becomes a monotonically increasing function of $p_i$. Since factor $p_i$ is more informative, in the sense that it reflects changes in both $\lambda$ and $\mu$, we suggest using it as a metric of channel quality, i.e.,

$$Q_i \equiv p_i \tag{39}$$

Next, we adopt a channel quality metric in order to be consistent with the residual energy metric:

$$W_{k,c} = \frac{|C_k^*| \sum_{j \in N_k^*} Q_j}{\sum_{i \in N_k^* \cup k} |C_i^*| \sum_{j \in N_i^*} Q_j} \tag{40}$$

Thus, Equation (3) is calculated taking into account (39) and (40).

*4.2. Limited Cluster Size*

Let us use queuing model $M/M/1$ to analyze the delay in CS packets. A similar approach was used in [20]. Let the cluster size be $n$. Cluster members' packets form a Poisson process with rate $n\lambda_{CS}$, where $\lambda_{CS}$ is the contribution of one CS. A cluster head serves packets, including its own. Service times of packets are assumed to be independent, exponentially distributed, random variables with mean $1/\mu$. The system's measures of effectiveness are well-known [21]. The packet latency is as follows:

$$T_{CS}(n, \mu, \lambda_{CS}) = \frac{1}{\mu - n\lambda_{CS}}, \ \ \mu > n\lambda_{CS} \tag{41}$$

Let $t_{QoS}$ be a strict packet delay constraint specified by a QoS policy, i.e., it is required that

$$T_{CS} \leq t_{QoS} \tag{42}$$

Assume that $\lambda_{CS}$ is large enough:

$$\lambda_{CS} > \frac{1}{n}\left(\mu - \frac{1}{t_{QoS}}\right) \tag{43}$$

In this case, inequality (42) is violated. In order to meet the QoS requirements, the number of cluster members can be reduced. Let us define a novel cluster size ($\hat{n}$) as follows:

$$\hat{n} = n(1 - \sigma), \ \ 0 \leq \sigma \leq 1 - \frac{1}{n} \tag{44}$$

where $\sigma$ is the fraction by which the cluster size is reduced. Therefore, the packet latency becomes

$$T_{CS}(\hat{n}, \mu, \lambda_{CS}) = \frac{1}{\mu - n(1 - \sigma)\lambda_{CS}} \tag{45}$$

In addition, let us consider the following relative difference:

$$\Delta = \frac{T_{CS}(n, \mu, \lambda_{CS}) - T_{CS}(\hat{n}, \mu, \lambda_{CS})}{T_{CS}(n, \mu, \lambda_{CS})} * 100\%, \ \ n > \hat{n} \tag{46}$$

We may rewrite this as

$$\Delta = \frac{\sigma\rho}{\sigma\rho + 1 - \rho} \times 100\% \tag{47}$$

here, $\rho$ is the utilization factor, i.e., it is the ratio of the rate at which cluster members generate packets to the rate at which the cluster head can handle this workload:

$$\rho = \frac{n\,\lambda_{CS}}{\mu} \tag{48}$$

As shown in Figure 6, QoS parameters can essentially be improved if the cluster size is reduced. Figure 6a shows the dependence of packet latency, $T_{CS}(\hat{n}, \mu, \lambda_{CS})$, on the fraction of the cluster reduction, $\sigma$, for different values of the initial cluster size, $n$. Here $\lambda_{CS} = 1$, $\mu = 50$. The contribution of $\mu$ is shown in Figure 6b, where $\lambda_{CS} = 1$, $n = 20$. Figure 6c illustrates packet latency versus $\sigma$ for various values of $\lambda_{CS}$. Here, $n = 10$, $\mu = 33$. In Figure 6d, we plot the relative difference, $\Delta$, as a function of the utilization factor, $\rho$, and $\sigma$.

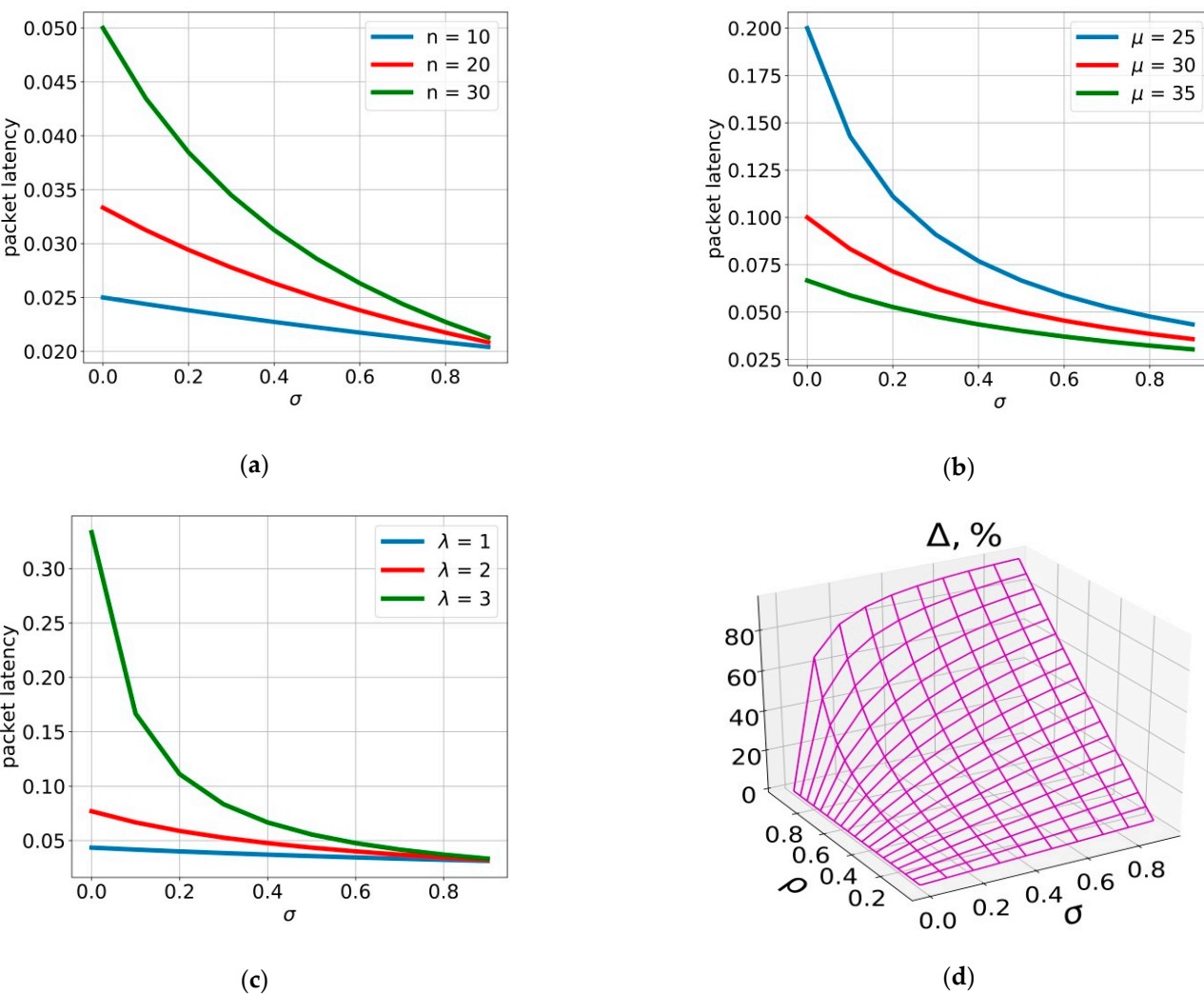

**Figure 6.** Packet latency versus cluster size reduction: (**a**) the impact of the initial cluster size; (**b**) the impact of cluster head performance; (**c**) the impact of traffic intensity from one cluster node; and (**d**) a 3D plot of the relative difference.

At the same time, the number of clusters is positively correlated with the cluster size. The system overhead increases with the number of clusters. It makes sense to use the

smallest appropriate number of clusters [22]. Therefore, the optimal cluster size is defined as follows:

$$n^* = \operatorname{argmax}\{k \in \mathbb{N} \mid T_{CS}(k) \le t_{QoS}\} \tag{49}$$

Using the floor function, we obtain

$$n^* = \left\lfloor \frac{1}{\lambda_{CS}} \left( \mu - \frac{1}{t_{QoS}} \right) \right\rfloor \tag{50}$$

Depending on the network topology, the number of neighboring nodes may be less than $n^*$. From this point of view, Formula (50) determines the maximal cluster size that guarantees QoS. It is a step function, an example of which is shown in Figure 7, where $\lambda_{CS} = 2$, $\mu = 21$.

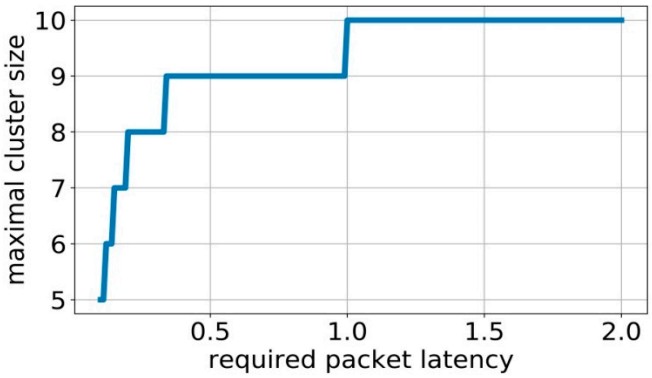

**Figure 7.** The cluster size's allowed limit versus the packet latency defined by the quality of service (QoS) policy.

Thus, unlike the NSAC protocol, we limit the cluster formation process to $n^*$ cluster members. This allows for the required packet latency.

Please note that our proposition reduces the re-clustering frequency as well. To address this point, we use an approach offered in [23]. Let us consider a network with a topology described by complete graph $K_n$. All $n$ sensors share the same licensed channels. The NSAC protocol creates one cluster of size $n$. The cluster head lifetime is as follows:

$$T_0 = \frac{E_B}{n \lambda_{CS} e_p} \tag{51}$$

where $E_B$ is the charge capacity of a CS battery, and $e_p$ is the energy consumption for transmitting one packet. In accordance with our proposition, the cluster size is defined by Formula (50), and the cluster head lifetime becomes

$$T_1 = \frac{E_B}{\left\lfloor \frac{1}{\lambda_{CS}} \left( \mu - \frac{1}{t_{QoS}} \right) \right\rfloor \lambda_{CS} e_p} \tag{52}$$

For example, if $\lambda_{CS} = 2$, $\mu = 11$, $t_{QoS} = 1$, then $n^* = 5$. Let $E_B = 6$ J, $e_p = 0.01$ J. The performance comparison of our approach and NSAC is shown in Figure 8.

Finally, note that the limited cluster size increases the passive resistance of networks to intrusions such as Black/Grey Holes.

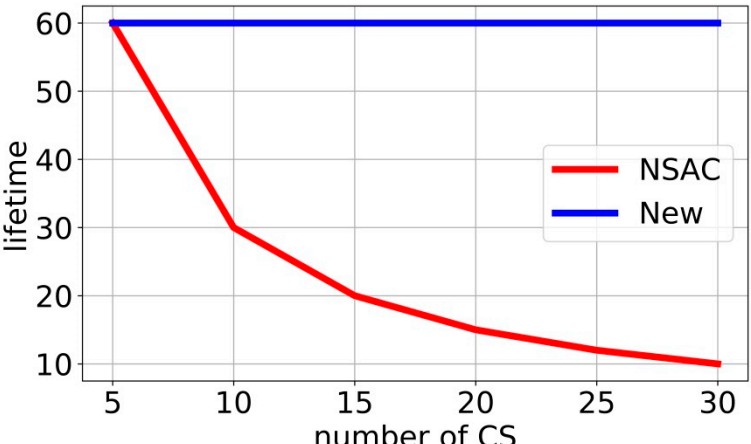

**Figure 8.** The performance comparison of the proposed method and network stability-aware clustering (NSAC).

## 5. Conclusions

Different from traditional clustering protocols, the recently proposed NSAC protocol considers both energy consumption and spectrum dynamics. It has been shown that that the proposed NSAC protocol clearly outperforms existing methods in the aspects of network stability and energy consumption. [11]. However, as we argued above, overall, NSAC fails to properly work. We proposed changing the channel quality metric and the method for calculating the weight of a CS. When we use the appropriate mathematical tools, we find there is no reason to consider the average idle duration and the probability of channel availability as competing entities. Therefore, we suggest using only the probability of channel availability as a metric of channel quality since this factor is more informative. The overall impact of the various channel states indicators on clustering protocol performance and reliability will be studied in future work.

**Author Contributions:** Conceptualization, V.S. and I.K.; methodology, V.S.; software, V.S.; validation, V.S. and I.K.; formal analysis, V.S.; investigation, V.S.; resources, V.S. and I.K.; data curation, V.S.; writing—original draft preparation, V.S.; writing—review and editing, V.S. and I.K.; visualization, V.S.; supervision, V.S. and I.K.; project administration, V.S. and I.K.; funding acquisition, V.S. and I.K. All authors have read and agreed to the published version of the manuscript.

**Funding:** This work was supported by a National Research Foundation of Korea (NRF) grant through the Korean Government (MSIT) under Grant NRF-2020R1I1A1A01065692.

**Data Availability Statement:** Data sharing is not applicable to this article.

**Conflicts of Interest:** The authors declare they have no conflict of interest.

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
