# Peer review of "An Efficient Clustering Protocol for Cognitive Radio Sensor Networks"

_electronics, doi:10.3390/electronics10010084_

Round 1
Reviewer 1 Report
This paper requires a lot of reworking in order to be admitted. Although your proposal seems to correct and improve a previous solution (the NSAC protocol), the way the paper is written make it difficult to understand.
Firstly, the English should be proofread. Some parts of the paper are unintelligible.
Secondly, the abstract and introduction should clearly expose the goal of your paper. Currently, they are confusing. For example, you say "In this paper we consider lacks of previously proposed clustering protocols.", when you only deal with one protocol (the NSAC).
The preliminaries section seems more a description of the NSAC protocol. State it at the beginning of the section.
Section 3, is a rebuttal of the NSAC protocol. Then, in section 4 you propose an improvement (or correction?) over this protocol using a Markov model, and includes some testing. No information is provided about how the results of figure 6, 7 and 8 are obtained.
It would necessary to describe precisely how the evaluation was performed including more results. I recommend creating a new section with all these results.
Author Response
1) To reflect the reviewer’s comment, the paper has been revised by professional English correction service.
2) To reflect the reviewer's comment, we modified and expanded the Introduction. The mentioned sentence has been changed as well.
3) To reflect the reviewer's comment, we organized subsections.
4) To reflect the reviewer’s comment, we have accompanied Section 4 with a preamble. The text readability has been improved by professional English native speaker. The computed values associated with the figures have been explicitly specified.
Reviewer 2 Report
The authors present work on clustering protocol in cognitive radio sensor networks. It is an interesting topic, but the presented work is about a too big research question while the presented work lacks novelty and technical depth. In addition, the paper is not well written.
- The authors "demonstrate that existing solutions do not operate properly ..." . I can't see a comprehensive review of the state of the art in details to obtain the conclusion accordingly.
- The authors "argue that the NSAC protocol fails to provide the declared goals". Is NSAC something fixed or a technical solution being investigated which we should compare with the state of the art?
- Proposals and experimental evaluation are not competitive.
- References are not correctly given following the format.
Author Response
To reflect the reviewer’s comment,
the text readability has been improved by professional English native speaker;
we modified and expanded the Introduction;
Section 4 is supplemented with an explanatory preamble;
the references format has been fixed.
Reviewer 3 Report
The submission presents an outstanding research that reveals the difficulties on state-of-the-art clustering protocols for wireless sensor networking when operating with remaining energy and quality of available common channels, even if their fusion is declared. Based on this, the authors discuss potential improvements for mitigating the targeted issues. Overall the paper is well written and presents high scientifically soundness. Although the covered research areas and challenges are widely present in the state-of-the art, this reviewer considers that the submission may be of the reader interest. Before acceptation, the following enhancements towards emphasizing its core contributions should be addressed:
- The Introduction does not properly highlight the motivation and core contributions of the presented research. In this regard, this reviewer suggests its extension by providing a clearer problem statement and enumerate the main contributions and differentiating aspects on the state of the art.
- Section 4 may include the principal/secondary goals of the conducted research (null/alternative hypothesis), assumptions, and limitations (the latter may be items to be researched/addressed at future work)
- A discussion section recapping the main identified issues on state-of-the-art solutions and the proposed mitigation actions should be included. Additionally, the conclusions section should be included at least by suggesting next related research steps.
Author Response
Thanks to the reviewers’ valuable comments that help improve paper quality. The paper has been extensively revised according to the reviewers’ comments.
1) To reflect the reviewer's comment, we modified and expanded the Introduction. The full text of the paper has been revised by professional English native speaker as well.
2) To reflect the reviewer’s comment, we have accompanied Section 4 with a preamble.
3) To reflect the reviewer’s comment, we expand the conclusions.
Round 2
Reviewer 1 Report
The authors address all my concerns. Congrats!
Author Response
Thanks to the reviewers’ valuable comments that help improve paper quality.
Reviewer 2 Report
The authors have made revisions which have covered all my concerns.
The authors need to make further improvement on the language and format to satisfy the requirements of this journal.
For instance, in page 3, line 124, the equation needs an number (2).
Author Response
Thanks to the reviewers’ valuable comments that help improve paper quality. To reflect the reviewer’s comment, we have numbered the equations and make the corresponding changes in the text.